# Enhanced Response for Foodborne Pathogens Detection by Au Nanoparticles Decorated ZnO Nanosheets Gas Sensor

**DOI:** 10.3390/bios12100803

**Published:** 2022-09-28

**Authors:** Cheng Zhao, Shanshan Xu, Jing Wei, Siqi Xie, Jinlei Wei, Jingting Han, Zhaohuan Zhang, Haiquan Liu, Jinsheng Cheng, Yong Zhao, Yongheng Zhu

**Affiliations:** 1College of Food Science and Technology, Shanghai Ocean University, Shanghai 201306, China; 2State Key Laboratory of Chemical Engineering, College of Chemical and Biological Engineering, Zhejiang University, Hangzhou 310027, China; 3The Key Laboratory of Biomedical Information Engineering of Ministry of Education, Institute of Analytical Chemistry and Instrument for Life Science, School of Life Science and Technology, Xi’an Jiaotong University, Xi’an 710049, China; 4Henry-Fork School of Food Sciences, Shaoguan University, Shaoguan 512005, China

**Keywords:** Au nanoparticles, ZnO nanosheets, MEMS gas sensor, 3-hydroxy-2-butanone, foodborne-pathogens detection, food safety

## Abstract

*Listeria monocytogenes* is a hazardous foodborne pathogen that is able to cause acute meningitis, encephalitis, and sepsis to humans. The efficient detection of 3-hydroxy-2-butanone, which has been verified as a biomarker for the exhalation of *Listeria monocytogenes*, can feasibly evaluate whether the bacteria are contained in food. Herein, we developed an outstanding 3-hydroxy-2-butanone gas sensor based on the microelectromechanical systems using Au/ZnO NS as a sensing material. In this work, ZnO nanosheets were synthesized by a hydrothermal reaction, and Au nanoparticles (~5.5 nm) were prepared via an oleylamine reduction method. Then, an ultrasonic treatment was carried out to modified Au nanoparticles onto ZnO nanosheets. The XRD, BET, TEM, and XPS were used to characterize their morphology, microstructure, catalytic structure, specific surface area, and chemical composition. The response of the 1.0% Au/ZnO NS sensors vs. 25 ppm 3-hydroxy-2-butanone was up to 174.04 at 230 °C. Moreover, these sensors presented fast response/recovery time (6 s/7 s), great selectivity, and an outstanding limit of detection (lower than 0.5 ppm). This work is full of promise for developing a nondestructive, rapid and practical sensor, which would improve *Listeria monocytogenes* evaluation in foods.

## 1. Introduction

*Listeria monocytogenes* (*LM*) is a foodborne pathogen that is able to cause acute meningitis, encephalitis, and sepsis. It has been found in food samples and raw materials, such as dairy products, meat products, vegetables, and seafood [1,2]. Since *LM* has a high mortality rate (20–30%), it is necessary to establish a rapid and accurate detection method for the bacteria in food samples and the environment [3]. At present, the detection methods applied to *LM* mainly include traditional physiological and biochemical measurements, molecular biological measurements targeting the virulence factors or specific genes of *LM*, and immunological measurements based on specific bindings of antigens and antibodies [4]. These methods generally require a long processing time, professional operators, or expensive instruments [5]. Thus, it is difficult to achieve rapid, nondestructive, and real-time in situ detection of the bacteria. According to previous studies, 3-hydroxy-2-butanone (3H-2B) is one of the characteristic microbial volatile organic compounds (MVOCs) of *LM*, whose abundance exceeds 32% [6]. Meanwhile, there is an excellent linear relationship between the incubation time of *LM* and the concentration of 3H-2B [7]. Consequently, it is feasible to realize a nondestructive and rapid evaluation of *LM* by detecting 3H-2B.

The metal oxide semiconductor (MOS) gas sensors realize gas detection through the conductivity changes of the MOS materials caused by chemical reactions between materials and target gases. With their advantages of portability and being economically and environmentally friendly, they have quickly become the most practical and widely-used gas sensors [8]. Since Deng et al. firstly studied 3H-2B sensors using mesoporous WO_3_ as sensing materials in 2017 [7], a variety of MOS materials with different morphologies has been developed and successfully applied to detect 3H-2B, such as WO_3_ nanoflowers [9], SnO_2_-Al_2_O_3_ nanocables [10], ZnO-Al_2_O_3_ nanocables [11], and NiO nano cuboids [12]. However, these materials are not all satisfactory in terms of response/recovery time, sensitivity, limit of detection (LOD), and selectivity. As one of the common n-type semiconductors, ZnO has the properties of small particles, large exciton binding energy (60 meV), a wide band gap (3.4 eV), a large specific surface area, and a small size effect [13]. Moreover, changing the physical morphology of materials is an effective way to improve their performance. For example, ZnO nanosheets (NS) are considered significant gas-sensing materials due to their simple synthesis method and excellent properties compared with the original ZnO described above [14,15].

In addition to morphology control, noble metal modification is also one of the important means to improve the sensing performance of MOS materials [16,17,18,19]. Au is a typical noble metal and is often selected as a sensitizer to modify MOS because Au is able to decrease the chemical adsorption activation energy of the measured gases and accelerate the reaction speed of the material. Moreover, it can also form Schottky contact with the MOS, so carriers are transferred from the MOS to the noble metal, which increases the Schottky barrier. At the same time, Au promotes the adsorption of O_2_ molecules, transfers them to the material surface to form oxygen negative ions, increases the thickness of the electron depletion layer (EDL), improves the conductivity, and finally improves the sensitivity [20,21]. For instance, Wang et al. showed that at 340 °C, Au/ZnO nanowires had a fast recovery time, greater response, better selectivity, and lower LOD for benzene and toluene than pure ZnO nanowires [22]. Therefore, it is feasible to develop a new ZnO NS gas sensor modified by Au nanoparticles (NPs) to detect 3H-2B, which is important for realizing the nondestructive and rapid assessment of *LM* in situ.

In this research, we synthesized ZnO NS by hydrothermal reaction, and prepared Au NPs (~5.5 nm) via the oleylamine reduction method. Then, we developed an excellent 3-hydroxy-2-butanone gas sensor based on the microelectromechanical systems (MEMs) using Au/ZnO NS as the sensing material. After working temperature tests, decoration material analysis, and load proportion optimization, the best performance of 1.0% Au/ZnO sensors were further studied. The response of these sensors vs. 25 ppm 3H-2B was up to 174.04 at 230 °C. Moreover, the sensors presented fast response/recovery time (6 s/7 s), excellent LOD (lower than 0.5 ppm), a good linear relationship, and great selectivity. Moreover, we explored the sensing mechanism according to the surface depletion model and electric resistance analysis. Such an excellent sensor based on 1.0% Au/ZnO NS signifies the amazing application potential in real-time, in addition to the nondestructive and efficient detection of 3H-2B. Considering the relationship between 3H-2B and *LM*, it opens up the opportunity to effectively and conveniently evaluate *LM* in foods, which will make a tremendous contribution to worldwide food safety.

## 2. Materials and Methods

### 2.1. Materials

We acquired chloroauric acid trihydrate (HAuCl_4_·3H_2_O, ≥99.9%) and 3-hydroxy-2-butanone (CH_3_COCH(OH)CH_3_, analytical standard) from Sigma-Aldrich, Saint Louis, United States. We purchased N-hexane from Shanghai Acmec Biochemical Co. Ltd., Shanghai, China. We purchased zinc acetate (C_4_H_6_O_4_Zn, ≥99.99%), urea (CO(NH_2_)_2_, ≥99.9%), oleylamine (C_18_H_37_N, 80–90%), and other reagents for selective testing from Yasuda chemical (Jiangsu) Co. Ltd., Huaian, China. We did not refine reagents in this research.

### 2.2. Instruments

We determined the crystalline form by X-ray diffraction (XRD, X’Pert) with Ni-filtered Cu Kα radiation (40 kV, 40 mA, 1.54056 Å) in the range of 2θ = 20 to 80° at 25 °C. We applied transmission electron microscopy (TEM) and high-resolution TEM (HRTEM; JEOI JEM-2011) to determine the microstructure of materials at accelerating voltage of 200 kV. We analyzed the chemistry composition by X-ray photoelectron spectroscopy (XPS; PHI-5000C ESCA), operating with Al Kα radiation (hν = 1486.6 eV). We charge-corrected binding energy values to C 1s = 284.6 eV. We tested the specific surface area by the Brunauer-Emmett-Teller method (BET; ASAP 2460), applying N_2_ as adsorbate at 77 K.

### 2.3. Synthesis of ZnO NS

We prepared ZnO NS as in previous methods [23,24,25]. We added 7 g urea and 2 g zinc acetate to 80 mL deionized water, stirred for 45 min, transferred to 100 mL conical flask, and laid hermetically in an oven at 95 °C for 6 h. After natural cooling, we centrifuged the white precipitate, washed, then dried at 80 °C. We calcined precursors in a muffle oven at 300 °C for 2 h to gain ZnO NS. These chemical reactions during the synthesis are shown in the following chemical equations:CO(NH_2_)_2_ + 3H_2_O → CO ↑ + 2NH_4_OH(1)
NH_4_OH → NH_4_^+^ + OH^−^(2)
4Zn^2+^ + CO_2_ + 6OH^−^ + 2H_2_O → Zn_4_CO_3_(OH)_6_·H_2_O + 2H^+^(3)
Zn_4_CO_3_(OH)_6_·H_2_O → 4ZnO + CO_2_ ↑ + 4 H_2_O ↑(4)

### 2.4. Synthesis of Au NPs

We prepared Au NPs as in previous methods [26]. We transferred 20 mL oleylamine and 118 mg chloroauric acid trihydrate to a three-necked flask (50 mL) and heated to 120 °C in constant-temperature oil bath, magnetically stirring and refluxing for 15 min. Then, we increased the temperature of the solution to 200 °C for 15 min, and increased again to 240 °C for 20 min. Next, we gathered Au NPs by centrifuging, cleaning, and drying. Finally, we dispersed Au NPs in N-hexane for further preserving. We carried out similar methods to prepare Pd and Pt NPs.

### 2.5. Synthesis of Au/ZnO NS

We prepared Au/ZnO NS as in previous methods [27]. We dispersed 100 mg ZnO NS in 30 mL deionized water. Then, we added various amounts of Au NPs to these solutions. Afterward, we ultrasonically stirred the solutions for 2 h. We set the mass percent of Au in the materials as 0.5, 1.0, 1.5, and 2.0 wt.%. Finally, we marked the nanocomposites as 0.5% Au/ZnO NS, 1% Au/ZnO NS, 1.5% Au/ZnO NS, and 2% Au/ZnO NS, and dried at 60 °C. In order to compare with Au/ZnO NS, we decorated 1.0 wt.% Pd and Pt NPs on ZnO NS and marked as 1.0% Pd/ZnO NS and 1.0% Pt/ZnO NS.

### 2.6. Preparation of the Sensors

We prepared the MEMS sensors according to the previous research [27]. The sensors possessed an interdigital electrode and an integrated microheater (Appendix A). Specifically, the resistance changes can be observed by the interdigital electrode, and a stable working temperature can be provided by the microheater. First of all, we transferred the gas−sensitive material and deionized water into an agate mortar and polished for 5 min to obtain a paste. Then, we added a drop of the abovementioned sample to the Pt interdigital electrode. After drying in an oven, we aged the ZnO sensors at 160 °C.

## 3. Results

### 3.1. Materials Characterization

As presented in Figure 1, ZnO NS were prepared by the one-step hydrothermal method combined with calcination, in which zinc acetate was the zinc source. Then, the oleylamine reduction method was used to synthesize Au NPs. Next, Au NPs were decorated on ZnO NS by ultrasonic treatment, in which the mass percentages of the Au element in Au/ZnO NS were set as 0.5, 1.0, 1.5, and 2.0 wt.%.

The crystal structure and phase of pristine ZnO and 1.0% Au/ZnO NS were measured by XRD. According to Figure 2, the two peaks belonging to pristine ZnO NS and 1.0% Au/ZnO were indexed as hexagonal ZnO (JCPDS card no. 89-0510). The lattice constants of it were a = 3.25 × 10^−10^ m and c = 5.20 × 10^−10^ m. There were no other stray peaks except the peaks belonging to ZnO, which illustrated that ZnO NS as-prepared had excellent purity. Meanwhile, the characteristic patterns of Au were not presented in the peaks of 1.0% Au/ZnO NS because of the weak peak intensity and the low load rate of Au [28].

TEM was performed to determine the morphology of sensing materials. As shown in Figure 3a, ZnO NS with typical nanosheet characteristics had lengths of about 40–60 nm. Their edge was clear and their boundary with the background was obvious, indicating that they had very good crystallinity [29]. Figure 3b presents the lattice spacing of ZnO NS, which was 0.19 nm, corresponding to the hexagonal ZnO (0 0 2) plane [30]. Figure 3c,d show Au NPs with diameters around 5.5 nm are uniform in size with great dispersion and crystallinity. The interplanar spacing of Au NPs was 0.24 nm, corresponding to the Au (1 1 1) plane [31]. Figure 3e shows a number of Au NPs were loaded on ZnO NS. The lattice spacing of the ZnO (0 0 2) and Au (1 1 1) plane are clearly shown in the HRTEM image of 1.0% Au/ZnO NS (Figure 3f). This revealed that Au NPs had been favorably decorated on ZnO NS. Moreover, the STEM-EDS elemental mapping images in Figure 3g–i showed that Zn, O, and Au elements were uniformly distributed in ZnO NS, further proving the success of Au/ZnO NS synthesis.

The chemical states of ZnO NS and the 1.0% Au/ZnO NS surface were applicated by the XPS analysis. The full spectrum is displayed in Figure 4a, which shows the signal intensity of Zn 2p, O 1s, and C 1s. It further shows that as-prepared ZnO NS had very high purity. Moreover, in the spectra of 1.0% Au/ZnO NS (Figure 4b), two weak peaks at 87.6 eV and 83.9 eV were distinguished to Au 4f_7/2_ and Au 4f_5/2_, revealing the Au element was in the valence of Au^0^ [32,33]. The O 1s patterns exhibited in Figure 4c were resolved by a Gaussian function to three peaks at 529.6, 531.1, and 531.4 eV, which were characteristic peaks of adsorbed oxygen (O_ads_), defect oxygen (O_def_), and lattice oxygen (O_lat_) [34,35]. The gas-sensing properties of sensitive materials were significantly affected by the valence of oxygen on their surfaces [28,36]. Figure 4c,d show the ratio of O_ads_ and O_def_ in pure ZnO NS increased from 10.2% and 21.0% to 14.8% and 27.2% after Au NPs sensitizing. In general, O_lat_ made little contribution to the electron transfer of n-type semiconductors because it was difficult to react with the measured gases. However, O_def_ and O_ads_ were in activity, so they could take part in a redox reaction and significantly increase the main charge-carriers’ concentration [37]. Therefore, the modification of Au NPs significantly increased the proportion of oxygen that participated in the reaction on the surface of Au/ZnO NS. This change may improve the sensitivity of the materials [28].

The N_2_ adsorption–desorption isotherms and BJH pore size distribution of ZnO NS and 1.0% Au/ZnO NS are exhibited in Appendix A. Both the pristine ZnO NS and 1.0% Au/ZnO NS were consistent with the characteristic type IV adsorption isotherm due to the hysteresis loops shown in Appendix A. It proved that as-prepared materials had a mesoporous framework [38,39,40]. Compared with the specific surface area of ZnO NS (24.74 m^2^/g), the specific surface area of 1.0% Au/ZnO NS increased by 20.3% (29.77 m^2^/g). Thus, 1.0% Au/ZnO NS had more effective adsorption sites to further improve the sensitivity. Moreover, the results above also revealed that most Au NPs were uniformly distributed on ZnO NS [41,42].

### 3.2. Gas-Sensing Properties

As mentioned above, 3H-2B has a 32.2% abundance in the MVOCs of *LM*, and the concentration of it is linearly related with the number of *LM* [6,7]. Thus, it is a simple and efficient way to indirectly evaluate *LM* in foods by detecting 3H-2B, ensuring food safety. Based on this, the environmentally-friendly MEMS sensors with magnificent sensitivity, high throughput, and miniaturization were fabricated, the method of which was described in previous work [27] using as-synthesized materials for 3H-2B detection.

For all MOS gas sensors, the operating temperature is an important factor that significantly affects their sensitivity [43]. Therefore, the sensors were firstly analyzed at temperatures from 170 °C to 290 °C vs. 25 ppm 3H-2B. It must be noted that for the n-type MOS gas sensor and reducing target gases, the response is defined as the value of R_air_ divided by R_gas_ (see more details in Supplementary Material). The optimum operating temperature of the pure ZnO NS was 260 °C (Figure 5a), when the response reached about 25. Then, the sensitive properties of the sensors after being decorated by Au NPs were obviously improved. While 1.0% Au/ZnO NS had the highest response as 174.04, which was almost seven times that of ZnO NS, its optimum working temperature was reduced to 230 °C. Moreover, most sensors presented the best performance at 230 °C, so the optimum operating temperature was set to 230 °C for following tests. Figure 5b exhibits the dynamic responses of 1.0% Au/ZnO NS, 1.0% Pd/ZnO NS, 1.0% Pt/ZnO NS, and pure ZnO NS vs. 1–25 ppm 3H-2B. Similar to the above results, the 1.0% Au/ZnO NS sensors showed the greatest sensitivity. Moreover, the responses of all sensors increased with the injection quantity of 3H-2B from 1 to 25 ppm and decreased from 25 to 1 ppm. During the rise and fall of the response value, when the concentration of 3H-2B was the same, the responses were almost the same. This indicated that the sensors had outstanding reversibility and repeatability [44]. These characteristics were also observed in the dynamic responses of 0.5%, 1.0%, 1.5%, and 2% Au/ZnO NS to 1–25 ppm 3H-2B (Figure 5c). According to the results in Figure 5b,c, 1.0% Au/ZnO NS sensors are considered to be the most outstanding. Meanwhile, the Au/ZnO-based gas sensor also presented a good linear relationship with the 3H-2B concentration, in which the R^2^ values were all greater than 0.99 (Figure 5d). To explore the LOD of the sensors, the response of 1.0% Au/ZnO NS vs. 3H-2B at 0.5 ppm was measured. As shown in Appendix A, the response of the 1.0% Au/ZnO NS sensor was vivid and evident (close to five) despite the 3H-2B concentration being only 0.5 ppm. Generally, the limit of the *LM* concentration in foods cannot exceed 100 CFU/g^−1^. According to the previous study, when the concentration of *LM* was 100 CFU/g^−1^, the corresponding concentration of 3H-2B was about 2.5 ppm [7]. However, the response of 1.0% Au/ZnO NS sensors was nearly 20 towards 2.5 ppm 3H-2B, which was significant enough to evaluate whether the foods were safe.

However, the rapid response/recovery time is one of the essential characteristics for gas sensors. So, that of Au/ZnO NS and ZnO NS towards 25 ppm 3H-2B was assessed and exhibited in Figure 6a. The response/recovery times of Au/ZnO NS were between 4 s and 15 s, and that of ZnO NS was 7 s and 20 s, respectively. This revealed that the modification of Au NPs significantly improved the response/recovery speed of the materials. Although the response/recovery time (6 s and 7 s) was not the fastest, it was still rapid enough to satisfy the urgent need for 3H-2B detection in real time. The responses of both 1.0% Au/ZnO NS and ZnO NS sensors were almost the same after five repeats of the test (Figure 6b), indicating that the reversibility and repeatability of these sensors were considerable, as mentioned above. Moreover, Appendix A displayed the long-term stability of 0.5%, 1.0%, 1.5%, and 2.0% Au/ZnO NS sensors at 230 °C towards the concentration of 3H-2B at 10 ppm. The responses of these sensors fluctuated indistinctly when they were examined every 5 days in a month. This was enough to illustrate the good long-term stability of these sensors. Then, the selectivity and discrimination analyses of these sensors were carried out due to the necessity of selectivity and discrimination characteristics for all sensors [45,46]. For studying the selectivity, the responses of Au/ZnO NS and ZnO NS sensors to formaldehyde, methanol, ethanol, ammonia, benzaldehyde, acetone, butanedione, and 3H-2B at 230 °C were investigated (Figure 6c). Their concentrations were all 25 ppm. Obviously, the responses of Au/ZnO NS and ZnO NS sensors toward 3H-2B were much higher than the responses towards other gases. Thus, the selectivity of these sensors, especially 1.0% Au/ZnO sensors, was quite outstanding. Meanwhile, the chemical sensitization effect was proven to be the most critical factor of enhanced sensitivity because the ratio of Au NPs decorated on ZnO NS barely influenced the selectivity [20,28]. The results of the discrimination evaluation are shown in Figure 6d. Compared with the response of the 1.0% Au/ZnO NS sensor to 25 ppm 3H-2B, the variation of the responses did not exceed 10%, no matter what interference gases were mixed with 3H-2B. Therefore, the selectivity and discrimination of the 1.0% Au/ZnO NS sensor were considerable.

Based on all the assessments and data above, the 1.0% Au/ZnO NS sensor has exhibited excellent gas-sensing performance. Furthermore, comparisons of its properties with other MOS-based 3H-2B sensors in the previous research are presented in Figure 7 and Appendix A. The indicators for comparisons mainly included the optimum working temperature, response to 3H-2B, response/recovery time, and LOD. According to the comparison, the properties of the 1.0% Au/ZnO NS sensor were almost superior in every index compared with other recently reported sensors reported [5,7,8,9,10,11,36]. Thus, it verified that the 1.0% Au/ZnO NS sensor had amazing application potential for the rapid, high-sensitivity, and good-selectivity detection of 3H-2B. This was also the basis for the real-time, nondestructive, and efficient evaluation of *LM*.

### 3.3. Gas-Sensing Mechanism

The sensing mechanism of ZnO NS adheres to a surface depletion model because ZnO NS is a kind of n-type MOS [47]. The reactions between the MOS surface and gas environment are the critical factors for the resistance change. Once ZnO NS is exposed to air, O_2_ molecules are adsorbed to the surface of ZnO NS through capturing electrons from the conduction band. At the same time, oxygen molecules convert into various kinds of O_ads_ ions. The reactions during the procedure are shown in the following equations:O_2 gas_ → O_2 ads_(5)
O_2 ads_ + e^−^ → O_2_^−^ _ads_(6)
O_2_^−^ _ads_ + e^−^ → 2O^−^ _ads_(7)
O^−^ _ads_ + e^−^ → O^2−^ _ads_(8)

Meanwhile, the potential barriers and resistances of the materials are increasing due to the construction of thick EDL on the ZnO NS surface. However, when ZnO NS comes into contact with 3H-2B, active O_ads_ ions on the material’s surface oxidize 3H-2B, which returns electrons to ZnO NS. The reaction is shown in the following equation [7]:CH_3_CHOHCOCH_3_ + 10O^−^_ads_ → 2CO_2_ + 4H_2_O + 10e^−^(9)

The electrons returned to ZnO NS are able to decrease the potential barriers and resistance of ZnO NS, which is displayed in Figure 8a,b [37]. In Au/ZnO NS, Au NPs can reduce the chemical adsorption activation energy of 3H-2B, which speeds up the reaction between 3H-2B and the O_ads_ ions. At the same time, the spillover effect occurs. Au NPs improve the adsorption efficiency of oxygen molecules and transfer them to the Au/ZnO NS surface, leading to the increased concentration of O_ads_. Moreover, the Schottky junction can be formed between Au NPs and the material interface. Moreover, compared with the work function of ZnO (4.65 eV), that of Au (5.1 eV) is higher; therefore, Au NPs attract more electrons from the conduction band of ZnO. All these processes greatly increase the thickness of EDL and the resistance of Au/ZnO NS, which improve the sensitivity significantly [20,21,48]. Figure 8c shows that the resistance of 1.0% Au/ZnO NS in the air was much higher compared with the resistance of pristine ZnO NS. Then, the resistance of 1.0% Au/ZnO NS decreased lower than that of ZnO NS after the 3H-2B injection. This phenomenon provided favorable support for the above mechanism analysis.

## 4. Discussion and Conclusions

In this work, ZnO NS was synthesized by a hydrothermal reaction, and Au NPs (~5.5 nm) were prepared via the oleylamine reduction method. Then, an ultrasonic treatment was carried out to modified Au NPs and ZnO NS, which were represented as Au/ZnO NS. XRD, BET, TEM, and XPS were used to characterize their morphologies, microstructures, catalytic structures, specific surface areas, and chemical compositions. The efficient 3H-2B gas sensors based on the MEMS were constructed using Au/ZnO NS as the sensing material. The response of the sensors vs. 25 ppm 3H-2B was up to 174.04 at 230 °C. Moreover, the Au/ZnO NS sensors presented fast response/recovery times (6 s/7 s), low LOD (0.5 ppm), a good linear relationship, and great selectivity. The gas sensors based on Pd/ZnO NS, Pt/ZnO NS, and pure ZnO NS were further studied to make comparisons with Au/ZnO NS. The enhanced responses of the sensors are primarily attributed to the morphology and structure improvements of Au/ZnO NS, the spillover effect, and the work function difference. Thus, such an excellent sensor signifies amazing application potential for the real-time, nondestructive, and efficient detection of *LM*, which will make a tremendous contribution toward worldwide food safety.

## Figures and Tables

**Figure 1 biosensors-12-00803-f001:**
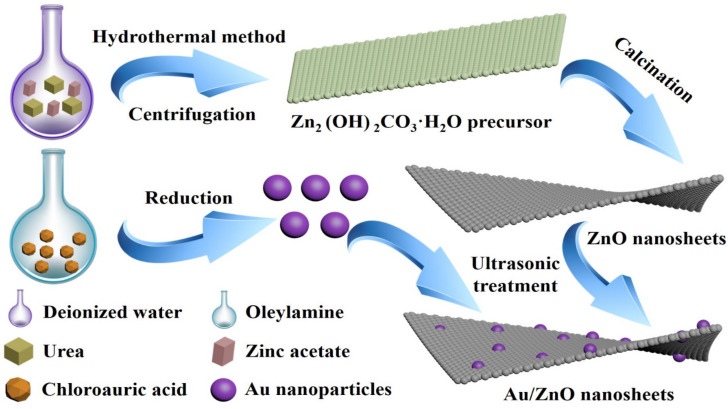
The synthesis schematic illustration of Au/ZnO NS.

**Figure 2 biosensors-12-00803-f002:**
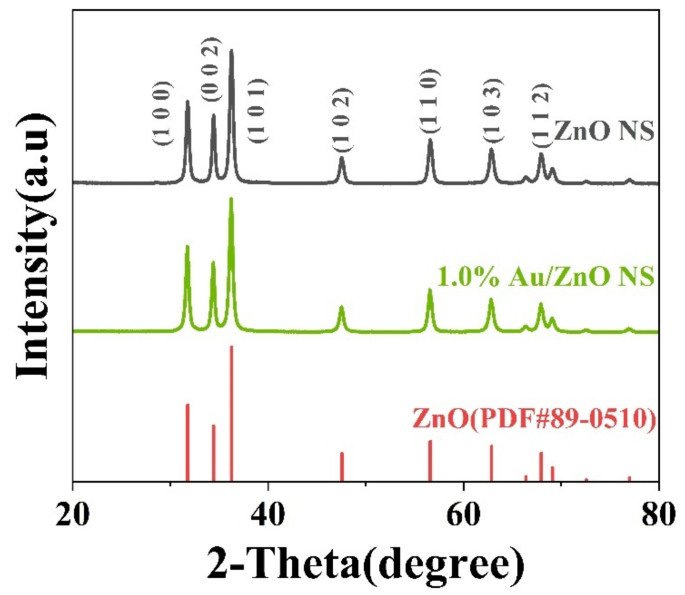
XRD patterns of ZnO NS and 1.0% Au/ZnO NS.

**Figure 3 biosensors-12-00803-f003:**
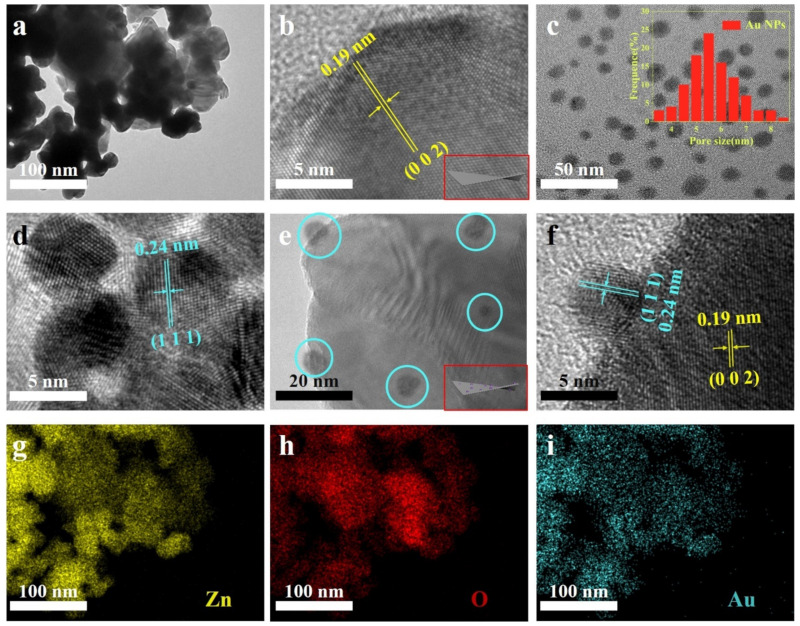
(**a**) TEM and (**b**) HRTEM images of pure ZnO NS. (**c**) TEM images and the particle size distribution of Au NPs. (**d**) HRTEM image of Au NPs. (**e**) TEM and (**f**) HRTEM images of 1.0% Au/ZnO NS. (**g**–**i**) STEM−EDS elemental mapping images of 1.0% Au/ZnO NS.

**Figure 4 biosensors-12-00803-f004:**
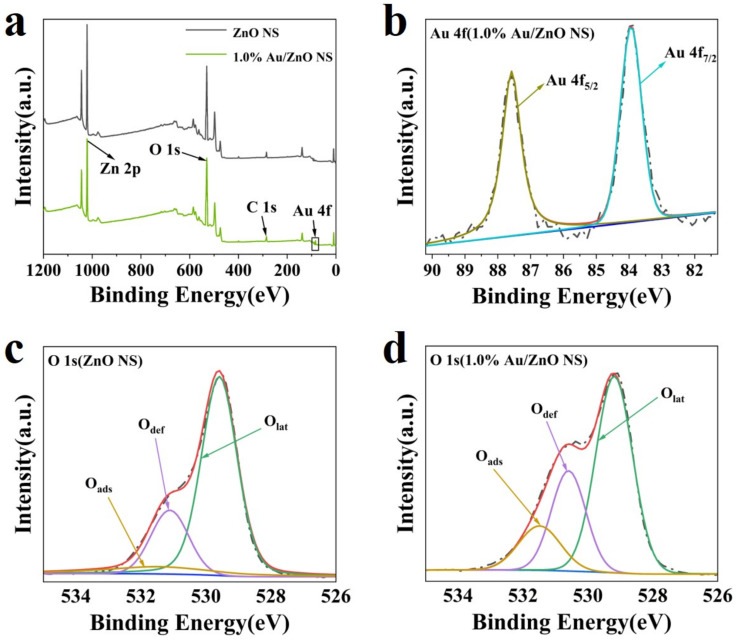
(**a**) Full XPS spectrum of ZnO NS and 1.0% Au/ZnO NS. (**b**) High-resolution Au 4f XPS spectra of 1.0% Au/ZnO NS. (**c**,**d**) High-resolution O 1s XPS spectra of ZnO NS and 1.0% Au/ZnO NS.

**Figure 5 biosensors-12-00803-f005:**
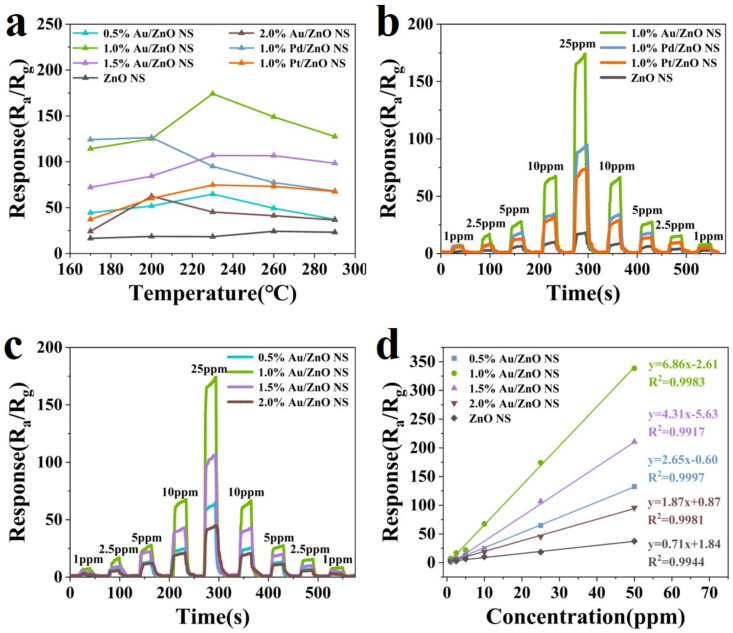
(**a**) Response of pure ZnO NS, 1.0% Pd/ZnO NS, 1.0% Pt/ZnO NS, and Au/ZnO NS to 25 ppm 3H-2B at working temperature from 170 °C to 290 °C. (**b**) Dynamic responses of 1.0% Au/ZnO NS, 1.0% Pd/ZnO NS, 1.0% Pt/ZnO NS, and pure ZnO NS to 1–25 ppm 3H-2B at 230 °C. (**c**) Dynamic responses of 0.5% Au/ZnO NS, 1.0% Au/ZnO NS, 1.5% Au/ZnO NS, and 2.0% Au/ZnO NS to 1–25 ppm 3H-2B at 230 °C. (**d**) Linear relationship between responses of 0.5% Au/ZnO NS, 1.0% Au/ZnO NS, 1.5% Au/ZnO NS, 2.0% Au/ZnO NS, and ZnO NS with 3H-2B concentrations (1–50 ppm) at 230 °C.

**Figure 6 biosensors-12-00803-f006:**
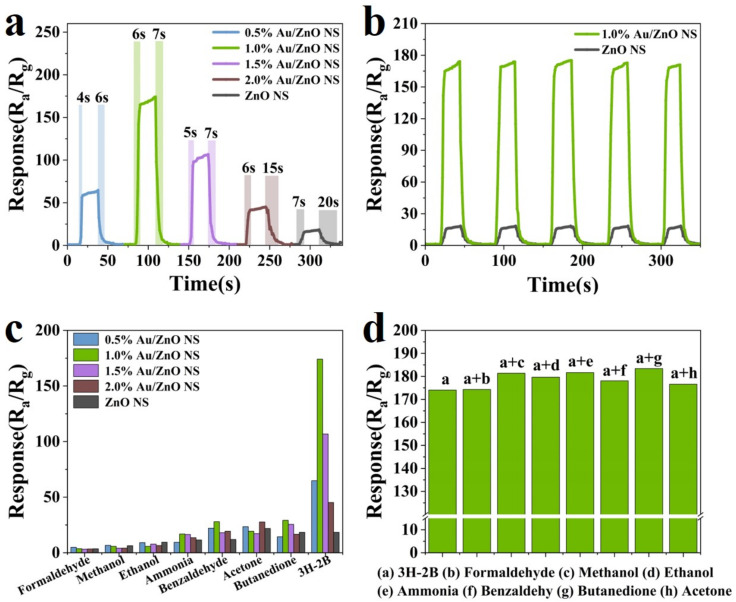
(**a**) The response/recovery time of 0.5% Au/ZnO NS, 1.0% Au/ZnO NS, 1.5% Au/ZnO NS, 2.0% Au/ZnO NS, and ZnO NS towards 25 ppm 3H-2B. (**b**) The repeatability of 1.0% Au/ZnO NS and ZnO NS to 3H-2B towards 25 ppm. (**c**) The selectivity of 0.5% Au/ZnO NS, 1.0% Au/ZnO NS, 1.5% Au/ZnO NS, 2.0% Au/ZnO NS, and ZnO NS towards 25 ppm interference gases and 3H-2B. (**d**) The discrimination test of 1.0% Au/ZnO NS sensors to the response of the mixed gas containing 25 ppm 3H-2B and other interference gases. All tests were operated at 230 °C.

**Figure 7 biosensors-12-00803-f007:**
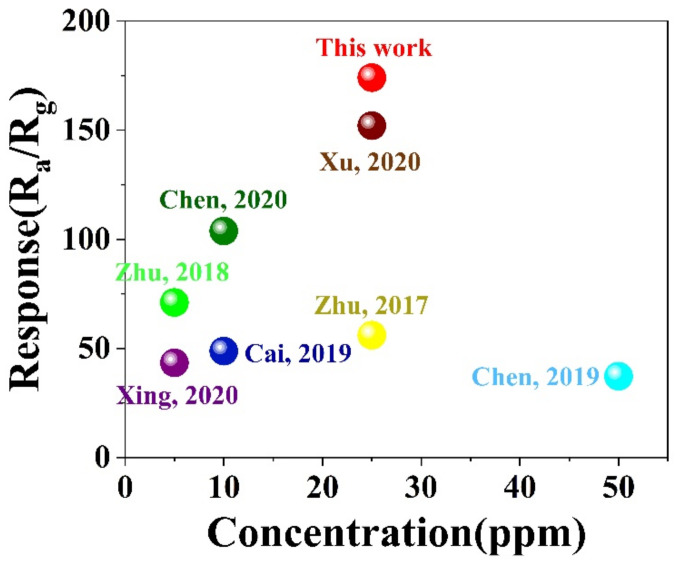
The comparison of the gas sensing performances of the 3H-2B sensors with other MOS-based 3H-2B sensors in previous research [5,7,8,9,10,11,36].

**Figure 8 biosensors-12-00803-f008:**
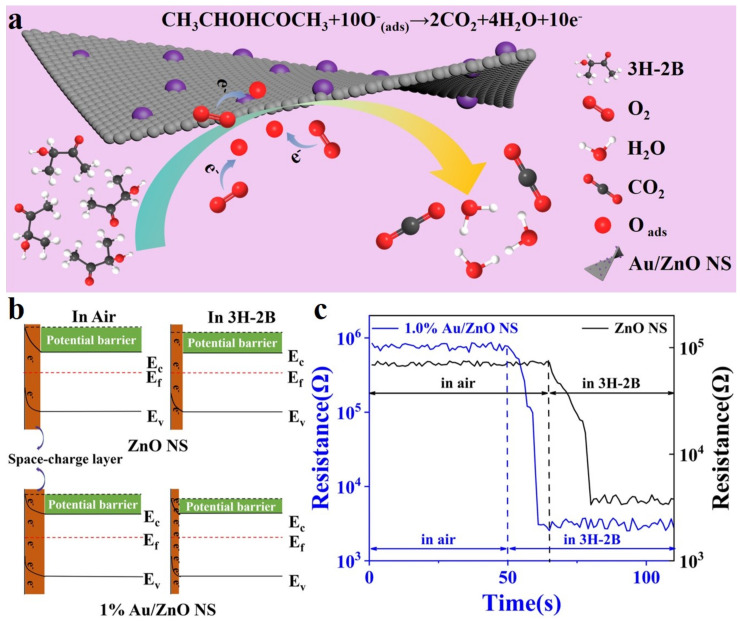
(**a**) The schematic illustration of 3H-2B sensing mechanism. (**b**) Electron structure change and (**c**) resistance change of ZnO NS and 1.0% Au/ZnO NS in air and 3H-2B.

## Data Availability

Not applicable.

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
