# Peer review of "Enhanced Response for Foodborne Pathogens Detection by Au Nanoparticles Decorated ZnO Nanosheets Gas Sensor"

_biosensors, 2022, doi:10.3390/bios12100803_

Round 1

Reviewer 1 Report

Authors present a gas sensor for the detection of foodborne pathogens, specifically 3-hydroxy-2-butanone, based on ZnO Nanosheet decorated with Au nanoparticles.

The manuscript is very well written and structured. Includes a very precise information about the experimental procedures and characterization. Regarding the gas sensor characterization is very satisfying to find an article dealing with cross sensitivity and selectivity, which is not usual.

For me, the paper can be published as it is with a couple of minor corrections:

1. It would be useful for the reader to know the expected concentration of the target analyte in that specific application, otherwise is rather difficul to know if the response of your sensor is good enough.

2. The "doping" concept is very specific and has a very narrow meaning and is not appropiate for your approach. You should name it like "Au NP decoration on ZnO..." or something similar.

Author Response

Biosensors

Manuscript ID:biosensors-1919264

Original Title: Enhanced Response for Foodborne Pathogens Detection by Au Nanoparticles Doped ZnO Nanosheets Gas Sensor

Revised Title: Enhanced Response for Foodborne Pathogens Detection by Au Nanoparticles Decorated ZnO Nanosheets Gas Sensor

Dear editors and reviewers,

Thank you very much for your review and constructive suggestions with regard to our manuscript “Enhanced Response for Foodborne Pathogens Detection by Au Nanoparticles Decorated ZnO Nanosheets Gas Sensor (Manuscript ID: biosensors-1919264)”. We carefully considered the reviewer’s comments and prepared a detailed, point-by-point, explanation of the revisions. The modifications in the revised manuscript have been marked using the “Track Changes” function for easy check purpose. On the following pages you will find our point-by-point response to editors and reviewers.

Thank you for considering our manuscript for publication in Biosensors.

If you have any questions about this revision, please feel free to contact me.

Best regards.

Yongheng Zhu, Professor

College of Food Science and Technology, Shanghai Ocean University, Shanghai, 201306

Response to Reviewer 1 Comments

General comments:

Authors present a gas sensor for the detection of foodborne pathogens, specifically 3-hydroxy-2-butanone, based on ZnO Nanosheet decorated with Au nanoparticles.

The manuscript is very well written and structured. Includes a very precise information about the experimental procedures and characterization. Regarding the gas sensor characterization is very satisfying to find an article dealing with cross sensitivity and selectivity, which is not usual.

Response: Thank you for considering our manuscript for publication in Biosensors. We also appreciate all your questions and suggestions.

Specific comments:

For me, the paper can be published as it is with a couple of minor corrections:

Point 1: It would be useful for the reader to know the expected concentration of the target analyte in that specific application, otherwise is rather difficul to know if the response of your sensor is good enough.

Response 1: Thanks a lot for your advice. In order to highlight the expected concentration of LM, we have added the following description in the revised manuscript (line 275 to 280):

    Generally, the limit of the LM concentration in foods cannot exceed 100 CFU/g-1. According to the previous study, when the concentration of LM was 100 CFU/g-1, the corresponding concentration of 3H-2B was about 2.5 ppm [7]. However, the response of 1.0% Au/ZnO NS sensors was nearly 20 towards 2.5 ppm 3H-2B, which was significant enough to evaluate whether the foods were safe.

Point 2: The "doping" concept is very specific and has a very narrow meaning and is not appropiate for your approach. You should name it like "Au NP decoration on ZnO..." or something similar.

Response 2: We really appreciate your suggestions. We have changed all “doped” and “doping” to “decorated” in the revised manuscript and supplementary materials.

Reviewer 2 Report

General comments:

 The manuscript presents a very good gas sensor for the biomarker 3-hydroxy-2-butanone (3H-2B). The sensor is based on Au-decorated ZnO nanosheets.

The authors did the characterization of the material, as well as the characterization of the sensor, studying its sensitivity and selectivity. Then, they compare the results obtained with the literature.

The discussion from line 192 to 201 is very good. The authors justify the reason why Au-decorated ZnO increases the material sensitivity, considering the increasing of “adsorbed” and “defects” Oxygen in the sensing material.

How about including a photograph of the MEMs sensor?

The results are great, but no information about the gas test methodology is presented. Did the authors use a setup already described in a previous work? Please indicate. How did the authors control the concentration of gases from 0.5 to 25 ppm? What is the volume of the gas test chamber? What was the gas flow rate through the chamber?

Specific comments:

 - Lines 23, 87 and 335: Replace “microelectron mechanical systems” with “microelectromechanical systems”.

- Line 54: The acronym “MOS” as “metal oxide semiconductor” refers to transistors (MOS-FET). In relation to gas sensors “MOS” stands for “metallic oxide sensor”. To avoid confusion, it is better use “MOX gas sensor” (metallic oxide gas sensor).

- Lines 84 to 94: I recommend that the authors rewrite this part. This is the end of the “Introduction”, but there are many methodology details and work conclusions. Authors should only indicate, in general terms, what the purpose of the work is.

- Figure 4: The “binding energy” axes are increasing in Fig. 4a, but decreasing in Figs. 4b, 4c and 4d. Please do all with the same orientation. In addition, Fig. 4d is clipped. “52” is actually “526”.

- Figure 6b: Please correct the caption into the figure:.The dark green curve corresponds to “ZnO NS”, but not to “Au/ZnO NS”.

- Figure 7: In this figure, temperature is not important. I suggest plotting the Response (Ra/Rg) vs. Concentration so that the reader can easily compare “this work” results with the literature results.

Author Response

Biosensors

Manuscript ID:biosensors-1919264

Original Title: Enhanced Response for Foodborne Pathogens Detection by Au Nanoparticles Doped ZnO Nanosheets Gas Sensor

Revised Title: Enhanced Response for Foodborne Pathogens Detection by Au Nanoparticles Decorated ZnO Nanosheets Gas Sensor

Dear editors and reviewers,

Thank you very much for your review and constructive suggestions with regard to our manuscript “Enhanced Response for Foodborne Pathogens Detection by Au Nanoparticles Decorated ZnO Nanosheets Gas Sensor (Manuscript ID: biosensors-1919264)”. We carefully considered the reviewer’s comments and prepared a detailed, point-by-point, explanation of the revisions. The modifications in the revised manuscript have been marked using the “Track Changes” function for easy check purpose. On the following pages you will find our point-by-point response to editors and reviewers.

Thank you for considering our manuscript for publication in Biosensors.

If you have any questions about this revision, please feel free to contact me.

Best regards.

Yongheng Zhu, Professor

College of Food Science and Technology, Shanghai Ocean University, Shanghai, 201306

Response to Reviewer 2 Comments

General comments:

The manuscript presents a very good gas sensor for the biomarker 3-hydroxy-2-butanone (3H-2B). The sensor is based on Au-decorated ZnO nanosheets.

The authors did the characterization of the material, as well as the characterization of the sensor, studying its sensitivity and selectivity. Then, they compare the results obtained with the literature.

The discussion from line 192 to 201 is very good. The authors justify the reason why Au-decorated ZnO increases the material sensitivity, considering the increasing of “adsorbed” and “defects” Oxygen in the sensing material.

How about including a photograph of the MEMs sensor?

The results are great, but no information about the gas test methodology is presented. Did the authors use a setup already described in a previous work? Please indicate. How did the authors control the concentration of gases from 0.5 to 25 ppm? What is the volume of the gas test chamber? What was the gas flow rate through the chamber?

Response: Thank you for considering our manuscript for publication in Biosensors. We also appreciate all your questions and suggestions. In response to your suggestions, we added the schematic illustration for the description of MEMS sensors (Figure S1). Besides, the preparation of ZnO sensors and gas sensor measurement section were increased in revised supplementary material as below (line 22 to 56):

Preparation of ZnO sensors

The microelectron mechanical systems (MEMS) sensor possesses interdigital electrode and an integrated micro heater, as shown in Figure S1. Specifically, the resistance changes can be observed by interdigital electrode, and stable working temperature can be provided by microheater.

The ZnO sensors was prepared according to the previous research [27]. First of all, the gas-sensitive material and ethanol were transferred into an agate mortar with polishing for a few minutes to obtain a paste. Secondly, a drop of above sample was added to the Pt interdigital electrode. After drying in infrared drying oven, the ZnO sensors were aged at 230 °C.

The testing process and the results were displayed through an intelligent analysis system. The working temperature of material chip was regulated by changing the heating voltage. During the test, an accurately calculated target liquid was injected onto a heated plate with a micro syringe, where the fan helped the injected liquid to vaporize rapidly. It can be visualized that the load resistance (RL) and the MEMS sensor formed a series circuit (Figure S1b), and at the same time the output voltage (Vout) could be recorded by software automatically.

Gas sensor measurement

The gas sensing performance of the gas sensor was measured by a static test system which records real-time change in resistance of sensor. The different concentrations of vapors were obtained by injecting liquid or gas of volume Q into a testing chamber. The volume Q can be determined by Eq 1:

                                                                                       (1)

Here, V, C, M, d, ρ, TR, and TB are the test chamber volume (90 mL), vapor concentration (ppm), molecular mass, liquid density, liquid purity, environmental temperature, and temperature in the testing chamber, respectively. 3-hydroxy-2-butanone (3H-2B) vapor was obtained by evaporating 8% 3H-2B solution. For the reducing gases and n-type MOS, the response of gas sensors is defined as S = Ra/Rg, where Ra is the gas sensor’s resistance in air atmosphere and Rg is the gas sensor’s resistance in target gas atmosphere. The response and recovery time are defined as the time taken by the gas sensor to achieve the resistance changes ranging from Ra to Ra-90% (Ra-Rg) and from Rg to Rg+90% (Ra-Rg) in the case of adsorption and desorption of target gases, respectively. And the relationship between the resistance (R) and the voltage (Vout) are listed in Eq2:

                                                                  (2)

In which, the circuit voltage (V) and the load resistance (RL) were set at 5 V and 100 KΩ, respectively. And at the same time the output voltage (Vout) could be recorded by software automatically.

Figure S1. Schematic illustration for the description of MOSs sensors. (a) The panoramic view of the MEMS gas sensor test system. (b) The measuring circuit of MEMS gas sensor. (c) The test base schematic of the MEMS gas sensor. (d) The exploded views of test base.

Reference (The order numbers of references are consistent with that in the Manuscript)

  1. Shen, J.B.; Xu, S.S.; Zhao, C.; Qiao, X.P.; Liu, H.Q.; Zhao, Y.; Wei, J.; Zhu, Y.H. Bimetallic Au@Pt nanocrystal sensitization mesoporous alpha-Fe2O3 hollow nanocubes for highly sensitive and rapid detection of fish freshness at low temperature. ACS Appl. Mater. Inter. 2021, 13, 57597-57608.

Specific comments:

Point 1: Lines 23, 87 and 335: Replace “microelectron mechanical systems” with “microelectromechanical systems”.

Response 1: Thank you very much for your advice. These problems have been fixed in the revised manuscript.

Point 2: Line 54: The acronym “MOS” as “metal oxide semiconductor” refers to transistors (MOS-FET). In relation to gas sensors “MOS” stands for “metallic oxide sensor”. To avoid confusion, it is better use “MOX gas sensor” (metallic oxide gas sensor).

Response 2: Thanks a lot for your suggestions. For this abbreviation, we consulted some experts and looked up more literatures in this field. Jung et al. reported “As far as it is known, this is the highest sensitivity achieved for p-type metal oxide semiconductor (MOS)-based gas sensors compared to previous studies” in the “Abstract” [1]. Mukherjee et al also described “Metal oxide semiconductors (MOS) are well known as reducing gas sensors” in the “Abstract” [2]. Besides, Aparicio et al. presented “This paper describes a novel approach to determine the individual contribution of volatile compounds to the overall sensor responses of metal oxide semiconductor (MOS) sensors when they are applied to a complex mixture of compounds such as food aroma” in 2010 [3]. According to these literatures, the abbreviation “MOS” is well understood and accepted to represent the meaning of “metal oxide semiconductor” in this field. Thus, We decided to keep the abbreviation unchanged.

Reference

  1. Choi, Y.M.; Cho, S.Y.; Jang, D.; Koh, H.J.; Choi, J.; Kim, C.H.; Jung, H.T. Ultrasensitive detection of VOCs using a high-resolution CuO/Cu2O/Ag nanopattern sensor. Adv. Funct. Mater. 2019, 29, 1808319.
  2. Ghosh, S.; RoyChaudhuri, C.; Bhattacharya, R.; Saha, H.; Mukherjee, N. Palladium silver-activated ZnO surface: Highly selective methane sensor at reasonably low operating temperature. ACS Appl. Mater. Inter. 2014, 6, 3879-3887.
  3. Garcia-Gonzalez, D.L.; Aparicio, R. Coupling MOS sensors and gas chromatography to interpret the sensor responses to complex food aroma: Application to virgin olive oil. Food Chem. 2010, 120, 572-579.

Point 3: Lines 84 to 94: I recommend that the authors rewrite this part. This is the end of the “Introduction”, but there are many methodology details and work conclusions. Authors should only indicate, in general terms, what the purpose of the work is.

Response 3: Thank you very much for your suggestion. We have updated the last paragraph of the “Introduction” part in the revised manuscript as below (Line 97 to 110):

In this research, ZnO NS were synthesized by hydrothermal reaction, and Au NPs (~ 5.5 nm) were prepared via oleylamine reduction method. Then, an excellent 3-hydroxy-2-butanone gas sensor developed based on the microelectromechanical systems (MEMs) using Au/ZnO NS as sensing material. After working temperature test, decoration material analysis and load proportion optimization, the best performance of 1.0% Au/ZnO sensors were further studied. The response of these sensors vs. 25 ppm 3H-2B was up to 174.04 at 230 °C. Moreover, the sensors presented fast response/recovery time (6 s/7 s), excellent LOD (lower than 0.5 ppm), good linear relationship and great selectivity. Besides, the sensing mechanism was explored according the surface depletion model and the electric resistance analyzation. Such an excellent sensor based on 1.0% Au/ZnO NS signifies the amazing application potential in real-time, nondestructive and efficient detection of 3H-2B. Considering the relationship between 3H-2B and LM, it opens up the opportunity to evaluate LM in foods effectively and conveniently, which will make the tremendous contribution for food safety in the whole world.

Point 4: Figure 4: The “binding energy” axes are increasing in Fig. 4a, but decreasing in Figs. 4b, 4c and 4d. Please do all with the same orientation. In addition, Fig. 4d is clipped. “52” is actually “526”.

Response 4: We really appreciate your correction. In the revised manuscript, we have adjusted the Figure 4 as below:

Figure 4. (a) Full XPS spectrum of ZnO NS and 1.0% Au/ZnO NS. (b) High-resolution Au 4f XPS spectra of 1.0% Au/ZnO NS. (c) and (d) High-resolution O 1s XPS spectra of ZnO NS and 1.0% Au/ZnO NS.

Point 5: Figure 6b: Please correct the caption into the figure:.The dark green curve corresponds to “ZnO NS”, but not to “Au/ZnO NS”.

Response 5: Thanks a lot for pointing out the mistake. It has been corrected in the revised manuscript as below:

Figure 6. (a) The response/recovery time of 0.5% Au/ZnO NS, 1.0% Au/ ZnO NS, 1.5% Au/ ZnO NS, 2.0% Au/ ZnO NS, and ZnO NS towards 25 ppm 3H-2B. (b) The repeatability of 1.0% Au/ZnO NS and ZnO NS to 3H-2B towards 25 ppm. (c) The selectivity of 0.5% Au/ ZnO NS, 1.0% Au/ ZnO NS, 1.5% Au/ ZnO NS, 2.0% Au/ ZnO NS, and ZnO NS towards 25 ppm interference gases and 3H-2B. (d) The discrimination test of 1.0% Au/ZnO NS sensors to the response of the mixed gas contained 25 ppm 3H-2B and other interference gases. All tests were operated at 230 °C.

Point 6: Figure 7: In this figure, temperature is not important. I suggest plotting the Response (Ra/Rg) vs. Concentration so that the reader can easily compare “this work” results with the literature results.

Response 6: Thank you very much for your suggestion. In the revised manuscript we have changed the Figure 7 to the comparison of the response with the concentration as below:

Figure 7. The comparison of the gas sensing performances of the 3H-2B sensors with other MOS-based 3H-2B sensors in the previous researches.
